# Can Nebulised Colistin Therapy Improve Outcomes in Critically Ill Children with Multi-Drug Resistant Gram-Negative Bacterial Pneumonia?

**DOI:** 10.3390/antibiotics8020040

**Published:** 2019-04-11

**Authors:** Fatih Aygun, Fatma Deniz Aygun, Fatih Varol, Cansu Durak, Haluk Cokugraş, Yildiz Camcioglu, Halit Cam

**Affiliations:** 1Department of Pediatric Intensive Care Unit, Istanbul University Cerrahpasa Medical Faculty, 34098 Fatih, Istanbul, Turkey; dr_fvarol@yahoo.com (F.V.); bzmrt@hotmail.com (C.D.); hacam@istanbul.edu.tr (H.C.); 2Department of Infectious Disease, Istanbul University Cerrahpasa Medical Faculty, 34098 Fatih, Istanbul, Turkey; fdenizaygun@gmail.com (F.D.A.); cokugras@gmail.com (H.C.); camciy@yahoo.com (Y.C.)

**Keywords:** colistin, nebulised, multiple drug resistance, pneumonia, *Pseudomonas*

## Abstract

In the past decade, multidrug-resistant (MDR) gram-negative bacteria have become a major problem, especially for patients in intensive care units. Recently, colistin became the last resort therapy for MDR gram-negative bacteria infections. However, nebulised colistin use was limited to adult patients. Thus, we investigated the efficacy and safety of nebulised colistin treatment against MDR microorganisms in the paediatric intensive care unit (PICU). Data of all patients admitted for various critical illnesses (January 2016 to January 2019) were reviewed. Differences between groups (with and without a history of nebulised colistin) were compared. Of 330 patients, 23 (6.97%) used nebulised colistin. Significant relationships were found between nebulised colistin usage and several prognostic factors (inotropic drug use (*p* = 0.009), non-invasive mechanical ventilation (*p* ≤ 0.001), duration in PICU (*p* ≤ 0.001), and C-reactive protein level (*p* = 0.003)). The most common microorganism in tracheal aspirate and sputum cultures was *Pseudomonas aeruginosa* (13 patients). The most common underlying diagnosis was cystic fibrosis, noted in 6 patients. No serious nephrotoxicity and neurotoxicity occurred. This study showed that colistin can be safely used directly in the airway of critically ill children. However, nebulised colistin use did not have a positive effect on mortality and prognosis.

## 1. Introduction

In the past decade, multidrug-resistant (MDR) gram-negative bacteria have become a major problem, especially among patients in intensive care units [1]. Recently, a World Health Organization (WHO) report showed that antimicrobial resistance was increasing worldwide [2]. The virulence of these MDR infectious agents severely restricts viable therapeutic options. Polymyxin E (colistin) was first used in the 1960s, and it has since been used for treating infections caused by gram-negative bacteria. However, in the 1970s, colistin use was discontinued because of drug-related nephrotoxicity and neurotoxicity [3]. Colistin was a last resort therapy for infections caused by MDR gram-negative bacteria, in particular *Pseudomonas aeruginosa*, *Acinetobacter baumannii*, and *Klebsiella pneumoniae* [3,4,5]. Most of the published experience of colistin as a treatment of pneumonia involved parenteral administration [5].

Nebulised antibiotics against MDR bacteria, developed as an alternative for intravenous therapy to deliver high concentrations of the antimicrobial agent directly to the site of infection, have the dual aims of improving efficacy and reducing toxicity [6]. The use of nebulised colistin was primarily limited to adult patients. Despite the lack of data on efficacy and safety from randomised controlled clinical trials, these agents have been recommended as viable therapeutic options for MDR nosocomial pneumonia and MDR ventilator-associated pneumonia in adults [6,7].

In this study, we aimed to investigate the efficacy and safety of nebulised colistin treatment for MDR microorganisms in the paediatric intensive care unit (PICU).

## 2. Results

The demographic characteristics of the 330 patients included in this study are shown in Table 1. Of these, 181 (54.8%) were male and 149 (45.2%) were female. Age distributions ranged from 2 days to 17.5 years with a median age of 2.0 years. The most frequent diagnoses were respiratory disorders (88 patients; 26.5%), metabolic diseases (51; 15.4%), neurological diseases (42; 12.7%), sepsis (43; 13.0%), and intoxication (21; 6.3%). The median duration of stay in PICU was 3.0 (2–122) days, while the mean Paediatric Risk of Mortality (PRISM)-III score was 9.66 ± 8.46. Overall, 23 (6.97%) patients used nebulised colistin. 

Among the patients who used nebulised colistin, 14 (60.9%) were female and 9 (39.1%) were male. In the nebulised colistin group, the mean age was 4.32 ± 3.9 years. There was a significant relationship between the use of nebulised colistin and prognostic factors, including inotropic drug use (*p* = 0.009), non-invasive mechanical ventilation (NIV) (*p* ≤ 0.001), PRISM-III score (*p* ≤ 0.001), duration in PICU (*p* ≤ 0.001), lymphocyte count (*p* ≤ 0.001), platelet count (*p* = 0.002), and C-reactive protein (CRP) level (*p* = 0.003) (Table 2). 

The most common microorganisms isolated from the tracheal aspirate and sputum cultures were *P. aeruginosa* (13 patients), *A. baumannii* (5 patients), *K. pneumoniae* (2 patients), *Serratia marcescens* (2 patients), and *Burkholderia cepacia* (1 patient). All microorganisms were resistant to two or more available systemic antibiotics (beta-lactams, quinolones, and aminoglycosides) that were tested, except colistin. In four patients, appropriate fungal treatment was commenced. The most common underlying diagnosis was cystic fibrosis in six (26.1%) patients. Eight patients subsequently developed bacteraemia after the pneumonia. The most commonly used beta-lactam antibiotic in addition to colistin was meropenem (19 patients). Five patients underwent tracheostomy and six patients had ventilator-associated pneumonia (Table 3).

The 61 patients who used colistin were divided into two groups: Combined colistin usage (colistin nebulised and intravenous usage) and only intravenous colistin usage. The number of patients with sepsis and continuous renal replacement therapy (CRRT) requirements was higher in the intravenous only group. *Pseudomonas* infection was more frequent in the combined group while *Klebsiella* growth was more frequent in the intravenous only group. There was no difference between the groups regarding acute kidney injury (AKI). There were no statistically significant differences between the groups for mortality (Table 4). 

A total of 104 patients were diagnosed with pneumonia during or after admission. These patients were divided into nebulised or non-nebulised colistin usage groups. There were no differences in the prognostic factors between these groups among patients with pneumonia. The duration of stay in the PICU was longer in the nebulised colistin group. Ventilator-associated pneumonia, concomitant sepsis, and high positive end expiratory pressure (PEEP) requirements were greater in the nebulised colistin group. Despite all these poor prognostic factors and MDR microorganisms in patients with nebulised colistin usage, no significant difference in outcomes was found between the two groups (Table 5).

## 3. Discussion

In this study, we aimed to show that nebulised colistin may confer added benefit to systemic colistin infusion for MDR infections in critically ill children. Overall, 23 (6.97%) patients were infected with MDR microorganisms, with *P. aeruginosa* and *A. baumannii* as the predominant infectious agents in tracheal aspirate and sputum cultures. As all microorganisms were colistin-susceptible, we used nebulised colistin when there was no treatment alternative for the MDR infections. The common feature among patients with colistin use was the persistence of pneumonia symptoms, despite the use of long-term antibiotics. A bacteriological and clinical response to nebulised colistin use was observed in 22 patients, and mortality only occurred in a female patient who had end-stage cystic fibrosis. There was no increase in AKI among the patients with nebulised colistin usage. Furthermore, there was no difference in mortality and prognostic factors in patients with nebulised colistin usage.

There are some underlying mechanisms to explain bacterial resistance against antimicrobial agents [8]. Multidrug efflux pumps are very important because they are a major cause of multiple drug resistance. The biological membrane of the bacteria can include multidrug efflux pump systems. Thus, the pump can extrude antibiotics from bacterial cells [8,9]. This mechanism is important in the development of antibacterial resistance for carbapenem [10]. The incidence of MDR among patients with gram-negative bacteria is increasing, and many antimicrobial agents are ineffective. Therefore, interest in colistin use has recently been rekindled [3,4,11]. In the literature, most uses of colistin for pneumonia involve parenteral administration. Increasing the dosage of intravenous colistin may help in achieving better concentrations in the lung. However, renal toxicity was reported to be a major adverse effect associated with intravenous colistin therapy [12,13]. Nebulised colistin against MDR microorganisms was developed as an alternative to intravenous therapy to deliver high concentrations of the antimicrobial agent directly to the infected lung [6,14]. Similarly, for a long time, nebulised aminoglycoside use has demonstrated that it has benefit in the treatment of cystic fibrosis [15].

In previous studies, combined treatment with colistin and other synergistic antibiotics showed better results for MDR infections, especially *A. baumannii.* Combined colistin and carbapenem, or tigecycline treatment, has been shown to provide synergistic effects against *A. baumannii* [16]. However, a recent study showed that efflux pumps play an important role in resistance to colistin [17]. Therefore, we did not use a single intravenous colistin treatment. We used the most commonly combined antibiotic therapies, including meropenem and colistin. Three patients were treated with nebulised colistin combined with tigecycline. In most patients who received meropenem with colistin, resistance to meropenem was reported. Our goal with this combination was to prevent the development of resistance against colistin. At first, there were no alternative antibiotics, other than colistin, for patients with MDR infection. Therefore, we commenced nebulised colistin in patients with similar clinical conditions to prevent possible antibiotic resistance. The treatment was continued in patients with no growth in the control culture following the commencement of colistin and after nebulised colistin was initiated.

The side effects of colistin are nephrotoxicity and neurotoxicity. The incidence rate of nephrotoxicity following colistin treatment was 14% to 24% [18]. In an adult study, a combination treatment with aminoglycosides showed that the combination may increase the nephrotoxic side effects of colistin [19]. Therefore, we administered colistin with close monitoring to determine the effect of colistin on patients’ kidney function. After discharge from the PICU, renal functions were monitored at the polyclinic. A meta-analysis showed that, when comparing the nebulised and intravenous colistin combination and the intravenous colistin alone for the treatment of ventilator-associated pneumonia, the combined treatment did not increase nephrotoxicity [10]. In contrast, the blood level of colistin was not required, as colistin does not cross the alveolar capillary barrier [14]; therefore, patients with chronic renal failure and impaired renal function due to multiple organ failure can also use nebulised colistin. However, we could not increase the dose of intravenous colistin due to nephrotoxicity in these patients. There was no serious occurrence of nephrotoxicity or neurotoxicity in our patients. The incidence of AKI did not increase, as seen in Table 2, Table 4, and Table 5. In addition, bronchospasm, hypersensitivity reactions, and fever did not occur in our patients. Therefore, it appears that nebulised colistin was well tolerated. In the patient with end-stage cystic fibrosis who was infected with MDR *P. aeruginosa*, clinical and laboratory findings did not resolve, and as a result she died. However, there was no significant difference in the eradication of MDR infections in the colistin groups. The majority of patients with nebulised colistin usage had pseudomonas infections, while in the IV colistin group, *Klebsiella* infections were predominantly noted. This difference between groups may be one of the factors affecting prognosis. Therefore, the use of nebulised colistin might be seen as ineffective. 

The need for mechanical ventilator support is a risk factor associated with poor prognosis in PICU patients [20]. We did not find a significant correlation between nebulised colistin and mechanical ventilator usage. However, a high positive end expiratory pressure (PEEP) requirement was associated with nebulised colistin treatment in pneumonia patients. During MDR infection, we preferred nebulised colistin for patients with a poor clinical picture and for those with pneumonia who showed no improvement while on standard antibiotic treatment. Therefore, nebulised colistin was used in more severe respiratory failure cases, and mortality was limited to only one patient. An important point here is that nebulised colistin did not contribute to respiratory function worsening. There was no need to increase the ventilator pressure during or after nebulised colistin usage. Although we had some concerns at first, we continued the treatment because we did not observe any negative side effects in nebulised colistin patients.

NIV was associated with nebulized colistin treatment. We suggest that this association is due to the post-extubation NIV usage that was more often in the nebulised colistin group. This correlation was not found in patients who used only colistin (Table 4).

In this study, there was a correlation between the length of PICU stay and nebulised colistin usage. This may be due to the more severe clinical outcomes in the nebulised colistin usage patient group and the slow response to treatment of patients with MDR infections. We found a significant relationship between nebulised colistin and increased CRP levels and PRISM scores. However, there were no differences regarding infection markers in patients with pneumonia with or without nebulised colistin usage. In patients with pneumonia, ventilator-associated pneumonia, concomitant sepsis, and high PEEP requirement were more frequent in the nebulised colistin group.

In our study, despite all of the negative prognostic factors in the combined colistin group, mortality only occurred in one patient as mentioned earlier, and the majority of MDR infections were completely eradicated. Overall, 17 (73.9%) patients’ nebulised colistin usage resulted in eradication of the MDR infectious agent. Although an improvement in clinical and laboratory parameters was observed in 22 patients, the control cultures continued to show replications in 5 patients. However, this was accepted as colonisation. Nebulised colistin treatment was maintained until there was no growth in the culture. In a study of adult patients who developed ventilator-associated pneumonia due to *Pseudomonas* and *Acinetobacter*, treatment success was reported in 85.7% [21]. In this previous study, the mortality rate (10/21) was found to be higher than in our study; however, we could not demonstrate that the use of nebulised colistin could provide a statistically significant decrease in mortality.

There are some limitations to our study. Our study is a retrospective and single-centre study. However, our study is the largest case series of nebulised colistin therapy in the paediatric population. 

In summary, there was no significant difference between CRP and leukocyte levels in the colistin groups. Only patients with inhaled colistin needed high PEEP. Nearly all our patients with nebulised colistin usage were cleared of the MDR infections, with an end to the colonisation. Because of this positive development, we continued treatment with nebulised colistin for MDR infections in our unit; although nebulised colistin usage decreased the positive culture results, no significant change was found on the prognosis in this study. There was no difference in mortality and prognostic factors according to nebulised colistin usage. Therefore, we consider that nebulised colistin might only be effective in correcting the culture results.

## 4. Materials and Methods

### 4.1. Study Design

Healthcare provision for children aged from 1 month to 18 years is provided in our PICU, which is equipped with 7 beds, 7 ventilators, and 2 isolation rooms. The data of all patients (*n* = 330) admitted to the PICU for various critical illnesses between January 2016 and January 2019 were extracted from electronic and written medical records (in accordance with the ethical principles for medical research), and included in this study. Patients with a history of a PICU stay duration of <24 h and those who died on the first day of admission were excluded. The study was conducted in accordance with the Declaration of Helsinki, and informed consent was obtained from parents or legal guardians of the patients when they were admitted to the PICU. The local ethics committee (İstanbul University-Cerrahpaşa, Ethical committee, no: 29430533-903.99-184025; 21 May 2018) approved the study. We recorded all materials, data, computer codes, and protocols associated with the publication for readers.

### 4.2. Patient Population and Data Collection

Before the commencement of treatment with nebulised colistin, all patients received prior treatment at least once with the appropriate antibiotics that were determined by the hospital antibiogram testing. In cases of continuing colonization or infection, nebulised colistin was commenced. Furthermore, patients with worsening clinical and radiological results also started nebulised colistin usage. Such patients had MDR microorganisms in their tracheal aspirate and sputum cultures. MDR is defined as acquired non-susceptibility to at least 1 agent in 3 or more antimicrobial categories [22]. Colistin susceptibility in these patients’ cultures was determined before the commencement of nebulised colistin.

The daily dosage of nebulised colistin was 5 mg/kg/day, which was divided into 2 doses, and colistin was diluted with sterile normal saline. The solution was nebulised through a micro-pump nebulizer (Aerogen, Dangan, Ireland). For intubated patients or after extubation, nebulised colistin was given by a conventional oxygen mask. Inhaled salbutamol was administered to all patients to prevent colistin-related bronchospasm, followed by respiratory physiotherapy. The intubation tube was not routinely replaced. The ventilator circuits were renewed weekly. All patients with MDR infections were isolated and the isolation rules were followed.

Demographic data and reason for hospitalisation were recorded. The patients’ sex, age, invasive or non-invasive mechanical ventilation requirement, duration of hospitalisation in the intensive care unit, mortality, PRISM-III score, continuous renal replacement therapy (CRRT), haemogram, and C-reactive protein (CRP) were recorded on admission. The patients were categorised into 2 groups based on a history or no history of nebulised colistin and the 2 groups were compared. Patients with colistin usage were divided into 2 groups according to combined (nebulised and intravenous colistin) and intravenous only colistin usage. The patients with pneumonia were divided into 2 groups with and without nebulised colistin usage. The relationship between the groups was evaluated for nebulised colistin usage.

AKI was defined as oliguria (urine output < 0.5 mL per kg of body weight per h) and an elevated serum creatinine value for the patient’s age, or a 1.5-fold increase in the baseline creatinine level in 24 h. 

PRISM score was calculated using the ‘https://kalite.saglik.gov.tr/’ website. The lowest/highest systolic blood pressure was recorded in the first 24 h for each patient, and diastolic blood pressure, heart rate and respiratory rate per minute, PaO_2_/FiO_2_, PaCO_2_, prothrombin time/partial thromboplastin time, serum total bilirubin, calcium, potassium, glucose, bicarbonate level, pupillary response, and Glasgow coma score were added to the relevant website for the PRISM score calculation.

### 4.3. Statistical Analysis

Statistical analysis was performed using the SPSS program (version 20.0, IBM Corporation, SPSS Inc., Chicago, IL, USA). Numerical data were expressed as mean ± standard deviation, while categorical data were expressed as frequency (n) and percentage (%). Comparisons of the differences in the baseline characteristics were performed using a Student’s t-test for parametric data and a Mann-Whitney U-test for non-parametric data. Categorical variables were compared using a chi-square test or Fisher’s exact test. A value of *p* < 0.05 was considered statistically significant.

## 5. Conclusions

This study showed that colistin could be safely used directly in the airway in critically ill children. In addition, nebulised colistin did not show any side effects. We believe that nebulised colistin can be a reasonable choice to minimise systemic side effects and increase the benefit of colistin therapy. However, despite MDR colonisation and inhibition of bacterial reproduction, nebulised colistin use did not have a positive effect on mortality and prognosis. Therefore, there is a need for prospective and multicentre studies involving larger numbers of patients on nebulised colistin usage.

## Figures and Tables

**Table 1 antibiotics-08-00040-t001:** Demographic characteristics of patients.

Number of Patients	*n* = 330
Sex Male Female	n (%) 181 (54.8%) 149 (45.2%)
Reasons for Hospitalisation Respiratory System Disease Metabolic diseases Neurologic Disease Sepsis Intoxication Cardiovascular Disease Haematology-oncology Other (post-operation etc.)	88 (26.5%) 51 (15.4%) 42 (12.7%) 43 (13.0%) 21 (6.3%) 12 (3.6%) 19 (5.7%) 56 (16.9%)
	Median (min–max)
Age	2.0 (2 days–17.5 years)
Duration of stay in the PICU	3.0 (2–122 days)
	Mean ± SD, n (%)
Acute kidney injury	52 (15.7%)
Inotropic medication	72 (21.7%)
CRRT	41 (12.3%)
Mechanical ventilation	99 (29.8%)
Death	23 (6.97%)
NIV	110 (33.1%)
PRISM-III score	9.66 ± 8.46
Nebulised colistin treatment	23 (6.97%)

PICU, Paediatric intensive care unit; CRRT, Continuous Renal Replacement Therapy; NIV, Non-Invasive Mechanical Ventilator; PRISM, Paediatric Risk of mortality; min, minimum; max, maximum; SD, standard deviation.

**Table 2 antibiotics-08-00040-t002:** Analysis of factors associated with adverse outcomes in patients with and without nebulised colistin usage.

Nebulised Colistin		Yes (*n* = 23)	No (*n* = 307)	*p*
**Sex**	Male Female	9 (39.1%) 14 (60.9%)	172 (56.0%) 135 (44.0%)	0.116
Age (years)		4.32 ± 3.89	5.24 ± 4.93	0.696
Mechanical ventilation		13 (56.5%)	138 (45.0%)	0.276
Inotropic drug use		10 (43.5%)	62 (20.2%)	0.009
Sepsis		4 (17.4%)	31 (10.1%)	0.200
Non-invasive mechanical ventilation		14 (60.9%)	96 (31.3%)	≤0.001
Blood component transfusion		13 (56.5%)	138 (45.0%)	0.276
PRİSM-III score		17.44 ± 14.30	11.44 ± 9.87	≤0.001
Leucocyte count (10^3^/µL)		20,077 ± 13,010	13,742 ± 6559	0.112
Lymphocytes count (10^3^/µL)		5288 ± 3846	2975 ± 2624	≤0.001
Platelet count (10^3^/µL)		439,682 ± 163,064	296,407 ± 183,269	0.002
C-reactive protein (mg/dL)		7.44 ± 10.35	3.05 ± 5.55	0.003
Sodium (mmol/L)		138.82 ± 4.56	139.30 ± 5.17	0.707
Chlorine (mmol/L)		99.56 ± 5.90	101.83 ± 6.12	0.149
Calcium (mg/dL)		9.44 ± 0.99	9.17 ± 1.03	0.305
Alanine aminotransferase (ALT) (U/L)		31.88 ± 56.63	107.95 ± 390.16	0.423
Aspartate aminotransferase (AST) (U/L)		46.47 ± 62.52	183.71 ± 772.24	0.465
Duration of stay in the PICU (days)		19.52 ± 16.83	7.00 ± 12.32	≤0.001
Mortality		1 (4.3%)	23 (7.5%)	0.580
CRRT		3 (13.0%)	38 (12.4%)	0.803
Acute kidney injury		3 (13.0%)	49 (15.9%)	0.927

PRİSM, Paediatric Risk of Mortality; PICU, Paediatric intensive care unit; CRRT, Continuous renal replacement therapy.

**Table 3 antibiotics-08-00040-t003:** Characteristics of the patients with nebulised colistin usage.

No	Sex	Age	Duration of Colistin Use(d)		Underlying Diseases	Isolated Pathogens	Bacteraemia	Tracheostomy	VAP	Eradication the Microorganism in the Cultures	Clinical and Radiological Improvement
			IV	NB							
1	F	15 y	14	12	Cystic fibrosis	*P. aeruginosa, C. albicans*	No	No	No	Yes	Yes
2	F	14 y	10	7	CVID, JIA	*P. aeruginosa, A. fumigatus*	Yes	No	No	Yes	Yes
3	F	4.5 y	25	20	Liver Transplant	*K. pneumoniae*	Yes	No	No	Yes	Yes
4	M	7 m	14	12	Pneumonia	*P. aeruginosa*	Yes	No	No	Yes	Yes
5	M	8 m	14	11	COPD	*P. aeruginosa, C. albicans*	No	Yes	No	No	Yes
6	F	14.5 y	16	14	Cystic fibrosis	*P. aeruginosa*	No	No	No	Yes	Yes
7	M	3 m	14	7	Pneumonia	*A. baumannii*	No	No	No	Yes	Yes
8	M	17 y	12	10	West, pneumonia	*P. aeruginosa, C. albicans*	No	No	No	Yes	Yes
9	F	18 m	40	30	SMA Type-I	*P. aeruginosa*	No	No	No	No	Yes
10	F	1 y	22	14	SMA Type-I	*A. baumannii*	No	Yes	No	No	Yes
11	M	17 y	21	7	CGH, COPD	*P. aeruginosa, A. fumigatus*	No	No	No	Yes	Yes
12	F	15 m	14	7	Cystic fibrosis	*S. marcescens*	Yes	No	No	Yes	No
13	M	2 m	28	22	MSUD, pneumonia	*K. pneumoniae*	Yes	No	No	Yes	Yes
14	F	2 y	11	11	SMA Type-I	*P. aeruginosa*	No	No	No	No	Yes
15	F	8 m	39	21	West Send.	*A. baumannii*	No	No	Yes	Yes	Yes
16	F	5 m	30	28	Cystic fibrosis	*P. aeruginosa, E. aerogenes*	Yes	No	No	Yes	Yes
17	F	3 m	42	36	CHD, hypotonic infant,	*A. baumannii*	Yes	No	Yes	Yes	Yes
18	F	17 y	21	8	Cystic fibrosis	*B. cepacia*	Yes	No	No	Yes	Yes
19	M	2.5 y	15	10	West send	*P. aeruginosa*	No	Yes	Yes	Yes	Yes
20	F	7 y	12	12	Cystic fibrosis	*P. aeruginosa*	No	No	No	No (died)	Yes
21	M	3 m	33	29	SMA	*P. aeruginosa*	No	Yes	No	Yes	Yes
22	F	4 m	35	15	Mitochondrial Disease	*A. baumannii*	No	Yes	Yes	Yes	Yes
23	F	6 m	21	13	Rhizomelic con.punc.	*P. aeruginosa*	No	No	Yes	No	Yes

VAP: ventilator associated pneumonia, CVID: common variable immune deficiency, JIA: juvenile idiopathic arthritis, COPD: chronic obstructive pulmonary disease, SMA: spinal muscular atrophy, MSUD: maple syrup urine disease, Rhizomelic con.punc.: Rhizomelic chondrodysplasia punctate, CHD: congenital heart defect.

**Table 4 antibiotics-08-00040-t004:** Analysis of factors associated with adverse outcomes in nebulised and intravenous colistin usage.

Type of Colistin Usage		Nebulised and Intravenous (*n* = 23)	Intravenous (*n* = 38)	*p*
Age (years)		4.32 ± 3.89	4.01 ± 4.43	0.800
Mechanical ventilation		13 (56.5%)	19 (50.0%)	0.152
Inotropic drug use		10 (43.5%)	19 (50.0%)	0.621
Non-invasive mechanical ventilation		14 (60.9%)	22 (57.9%)	0.147
Blood component transfusion		13 (56.5%)	32 (84.2%)	0.009
PRİSM-III score		17.44 ± 14.30	14.65 ± 10.56	0.162
Leucocyte count (10^3^/uL)		20,077 ± 13,010	9183 ± 5778	≤0.001
Lymphocytes count (10^3^/uL)		5288 ± 3846	2486 ± 2699	0.004
Platelet count (10^3^/uL)		439,682 ± 163,064	178,417 ± 147,683	≤0.001
C-reactive protein (mg/dL)		7.44 ± 10.35	5.69 ± 8.19	0.508
Sodium (mmol/L)		138.82 ± 4.56	140.25 ± 10.02	0.579
Chlorine (mmol/L)		99.56 ± 5.90	102.09 ± 10.76	0.385
Calcium (mg/dl)		9.44 ± 0.99	8.54 ± 0.89	0.002
Albumin level		3.74 ± 0.92	3.21 ± 0.71	0.043
Duration of stay in the PICU (days)		19.52 ± 16.83	14.71 ± 13.23	0.354
CRRT		3 (13.0%)	18 (47.4%)	0.016
Acute kidney injury		3 (13.0%)	13 (34.2%)	0.175
Sepsis		8 (34.8%)	25 (65.8%)	0.019
Mortality		1 (4.3%)	8 (21.1%)	0.075
Eradication the microorganism		18 (78.3%)	31 (75.2%)	0.752
PEEP > 10 cm H_2_0		12 (52.2%)	3 (7.9%)	≤0.001
Microorganism	*Klebsiella*	2 (8.7%)	20 (52.6%)	≤0.001
	*Pseudomonas*	13 (56.5%)	5 (13.2%)	
	*Acinetobacter*	5 (21.7%)	3 (7.9%)	
	Other	3 (13.0%)	10 (26.3%)	

PRİSM, Paediatric Risk of Mortality; PICU, Paediatric intensive care unit; CRRT, Continuous renal replacement therapy.

**Table 5 antibiotics-08-00040-t005:** Analysis of factors associated with adverse outcomes in patients with pneumonia with or without nebulised colistin usage.

Patients with Pneumonia in the PICU	Nebulised Colistin Usage		
	Yes (*n* = 23)	No (*n* = 81)	*p*
Mechanical ventilation	13 (56.5%)	33 (40.7%)	0.179
Duration of mechanical ventilation (days)	9.45 ± 10.57	12.75 ± 13.67	0.432
Inotropic drug use	10 (43.5%)	19 (23.5%)	0.059
Non-invasive mechanical ventilation	14 (60.9%)	47 (58.0%)	0.060
Blood component transfusion	14 (60.9%)	38 (46.9%)	0.237
Leucocyte count (10^3^/uL)	20,665 ± 13,163	17,766 ± 15,567	0.691
C-reactive protein (mg/dL)	7.44 ± 10.35	3.98 ± 6.26	0.074
Duration of stay in the PICU (days)	19.52 ± 16.83	9.98 ± 16.32	0.012
Acute kidney injury	4 (17.4%)	6 (7.4%)	0.059
Ventilator-associated pneumonia	6 (26.1%)	3 (3.7%)	≤0.001
Sepsis	8 (34.8%)	20 (24.7%)	≤0.001
PEEP > 10 cm H_2_0	12 (52.2%)	3 (3.7%)	≤0.001
Mortality	1 (4.3%)	3 (3.7%)	0.856

PICU, Paediatric intensive care unit; PEEP, Positive end expiratory pressure.

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
