# Peer review of "Can Nebulised Colistin Therapy Improve Outcomes in Critically Ill Children with Multi-Drug Resistant Gram-Negative Bacterial Pneumonia?"

_antibiotics, 2019, doi:10.3390/antibiotics8020040_

Round 1
Reviewer 1 Report
At present little is known of the clinical utility of nebulized colistin in the general patient population, and its use has been limited to patients with cystic fibrosis The aim of the present study could be important to improve treatments for multidrug resistant bacteria by introducing nebulized colistin. The authors investigate the efficacy and safety of nebulized colistin treatment in 23 children admitted to the ICU. However, this study lacks scientific rigor since there are some methodological errors and experimental biases that need to be clarified. In addition, I think the present study is not enough to demonstrate colistin is effective and it could be used safely.
In my opinion the main drawback of the paper concerns the study population. The study must include only those patients who received a treatment with colistin, or at least only those patients with respiratory infectious diseases. There is a large proportion of patients who did not receive the treatment, simply because they do not need a treatment with a specific antibiotic like colistin. This issue distorts and biases the results. In addition, the statistically significant results were more related to prognostic factors than to disease outcomes.
The definition of multi-drug resistance (MDR) must be revisited. Reference [7] in line 165 is not correct. MDR is defined as acquired non-susceptibility to at least one agent in three or more antimicrobial categories (See Magiorakos et al. 2011. Clin Microbiol Infect 2012; 18: 268–281). The susceptibility profiles of microorganisms isolated from the tracheal aspirate and sputum cultures should be shown, particularly colistin susceptibility.
Study design (methods) should explain better how to collect data, analyze and interpret them to show clearly this study is properly planned. This section should establish, among other things, the suitability of the type of study, the study population, and the calculation of sample size.
Title. “gram-negative bacterial pneumonia” instead of “gram-negative bacterial pneumonia”
In general, the work is not written in correct scientific English.
Author Response
01 March 2019
Prof. Dr. Christopher C. Butler
Editor-in-Chief
Antibiotics MDPI
Dear Prof. Butler:
I, along with my coauthors, would like to re-submit the attached manuscript entitled “Nebulised colistin therapy can improve outcomes in critically ill children with multi-drug resistant Gram-negative bacterial pneumonia” as an Original Research article. The manuscript ID is MS No.: antibiotics-451399
The manuscript has been carefully rechecked and appropriate changes have been made in accordance with the reviewers’ comments. The responses to the reviewers’ comments have also been prepared. We hope that the revised manuscript is now suitable for publication in your journal.
I look forward to your reply.
Sincerely,
Fatih Aygun
Department of Pediatric Intensive Care Unit
Istanbul University Cerrahpasa Medical Faculty, 34098, Fatih, Istanbul, Turkey
Tel.: +90 (532)532 786 86 82; (530)5534513
Fax: +90 (212) 6328633
Email: faygun9@hotmail.com
Note:
1. I changed the title
2. Abstract: It was edited it to achieve the word count below 200 words.
3. Keywords: The keywords to be added after the abstract were in alphabetical order.
4. The Introduction has been rechecked
5. Results: We rewritten the results. The changes made in the Results were essentially in the yellow-highlighted area.
6. We rewritten the discussion.
7. Materials and Methods: rechecked and rewritten.
8. Statistical analyses: No changes were required here.
9. Tables: Table 4 and 5 were added.

Reviewer 2 Report
The manuscript entitled "Nebulised colistin therapy can improve 2 outcomes in critically ill children with multi-drug resistant gram-negative pneumonia" is a retrospective study on nebulised colistin therapy in PICU patients. More data about colistin therapy are absolutely important, thus this manuscript potentially carries very useful data.
I do not understand if the control Group (n=307) is taking systemic colistin or not. For me this is a key Information that I cannot find reading the manuscript.
My understanding is that the control Group was not characterized by the use of colistin in any route of administration. On this understanding, I found extremely difficult to understand the rationale behind the Group selections. One Group (the colistin one) is characterized by multidrug resistant infections, the second Group simply collects all the other patients regardless of the disease. With such Groups the comparisons seem to me difficult to interpret. The aim of the study is "to Show that nebulised colistin may confer added benefit to systemic infusionj for MDR infections in critically ill children", but I cannot understand how the authors can draw the conclusion that "Nebulised colistin therapy can improve outcomes in critically ill children with multi-drug resistance gram-negative pneumonia.
Author Response

(The authors gave the same response as above.)

Round 2
Reviewer 1 Report
Based on new Table 5, what data supports the conclusion that colistin can improve outcomes in critically ill children (title)? I suggest modifying the title.Tables 4 and 5. Could it be possible to add data and statistical analysis about whether the treatment with colistin eliminated the isolated MDR microorganisms?
Suggestions for table titles. "...analysis of factors associated with adverse outcome in....." or "....risk factors in patients with..." instead of "Comparison of the patients...."
Line 38 Klebsiella pneumoniae.
Check bacterial names in Table 3 and italicize them
Tables 1, 2 and 3 could be supplementary material
Author Response
Dear Editor,
I, along with my coauthors, would like to re-submit the attached manuscript entitled “Can nebulised colistin therapy can improve outcomes in critically ill children with multi-drug resistant gram-negative bacterial pneumonia?” as an Original Research article. The manuscript ID is MS No.: Antibiotics-451399
The manuscript has been carefully rechecked and appropriate changes have been made in accordance with the reviewers’ comments. The newer changes are now with green highlights while previous revision is with yellow highlights. The responses to the reviewers’ comments have also been prepared. We hope that the revised manuscript is now suitable for publication in your journal.
I look forward to your reply.
Sincerely,

Reviewer 2 Report
I appreciated the new analyses provided by the authors. The revised manuscript entitled "Nebulised colistin therapy can improve outcomes in critically ill children with multi-drug resistant gram-negative pneumonia" provides a more comprhensive Picture as compared with the first Version. However it is still difficult to understand the message of this manuscript, especially because the authors did not properly discussed the new results. I found particularly informative table 4 and 5.
In table 4 the authors compared the effect of nebulised colistin on top of intravenous colistin. What the author would say about the main markers of infections? Is the intravenous dose comparable between the 2 Groups? This is also extremely important to clarify and may help to explain why the nebulised/intravenous Group has a lower incidence of AKI (albeit not significant)compared with the intravenous Group. Additionally, it seems to me that there is no benefit of adding nebulised colistin to intravenous colistin. I would argue that there is no point in administer nebulised colistin to patients that are already taking intravenous colistin. These are the most important aspects of this retrospective study and must be discussed properly. If possible add the PEEP values to table 4 as is reported in table 5.
In table 5 the authors stated that PPEP is significantly higher in the colistin Group, however all the other markers are not improved. This cannot be ignored and should be fairly discussed.
Author Response

(The authors gave the same response as above.)

Round 3
Reviewer 1 Report
No additional comments.
Author Response
Dear Reviewer:
I, along with my coauthors, would like to re-submit the attached manuscript entitled “Can nebulised colistin therapy can improve outcomes in critically ill children with multi-drug resistant gram-negative bacterial pneumonia?” as an Original Research article. The manuscript ID is MS No.: Antibiotics-451399
We have formatted our manuscript for submission to Antibiotics. The revisions are added with yellow highlights. Editing certificate also received and added.
I look forward to your reply.
Sincerely,
Fatih Aygun
Department of Pediatric Intensive Care Unit
Istanbul University Cerrahpasa Medical Faculty, 34098, Fatih, Istanbul, Turkey
Tel.: +90 (532)532 786 86 82; (530)5534513
Fax: +90 (212) 6328633
Email: faygun9@hotmail.com

Reviewer 2 Report
Thanks to the authors for the answers and modifications. Now the manuscript seems clear in its Overall Goal.
Author Response
Dear Editor:
I, along with my coauthors, would like to re-submit the attached manuscript entitled “Can nebulised colistin therapy can improve outcomes in critically ill children with multi-drug resistant gram-negative bacterial pneumonia?” as an Original Research article. The manuscript ID is MS No.: Antibiotics-451399
We have formatted our manuscript for submission to Antibiotics. The revisions are added with yellow highlights. Editing certificate also received and added.
I look forward to your reply.
Sincerely,
Fatih Aygun
Department of Pediatric Intensive Care Unit
Istanbul University Cerrahpasa Medical Faculty, 34098, Fatih, Istanbul, Turkey
Tel.: +90 (532)532 786 86 82; (530)5534513
Fax: +90 (212) 6328633
Email: faygun9@hotmail.com
